# Integration of Genomic and Clinical Retrospective Data to Predict Endometrioid Endometrial Cancer Recurrence

**DOI:** 10.3390/ijms232416014

**Published:** 2022-12-16

**Authors:** Jesus Gonzalez-Bosquet, Sofia Gabrilovich, Megan E. McDonald, Brian J. Smith, Kimberly K. Leslie, David D. Bender, Michael J. Goodheart, Eric Devor

**Affiliations:** 1Department of Obstetrics and Gynecology, University of Iowa, 200 Hawkins Dr., Iowa City, IA 52242, USA; 2Department of Biostatistics, University of Iowa, 145 N Riverside Dr., Iowa City, IA 52242, USA; 3Division of Molecular Medicine, Departments of Internal Medicine and Obstetrics and Gynecology, The University of New Mexico Comprehensive Cancer Center, 915 Camino de Salud, CRF 117, Albuquerque, NM 87131, USA

**Keywords:** endometrial cancer, recurrence, prediction, machine learning

## Abstract

Endometrial cancer (EC) incidence and mortality continues to rise. Molecular profiling of EC promises improvement of risk assessment and treatment selection. However, we still lack robust and accurate models to predict those at risk of failing treatment. The objective of this pilot study is to create models with clinical and genomic data that will discriminate patients with EC at risk of disease recurrence. We performed a pilot, retrospective, case–control study evaluating patients with EC, endometrioid type: 7 with recurrence of disease (cases), and 55 without (controls). RNA was extracted from frozen specimens and sequenced (RNAseq). Genomic features from RNAseq included transcriptome expression, genomic, and structural variation. Feature selection for variable reduction was performed with univariate ANOVA with cross-validation. Selected variables, informative for EC recurrence, were introduced in multivariate lasso regression models. Validation of models was performed in machine-learning platforms (ML) and independent datasets (TCGA). The best performing prediction models (out of >170) contained the same lncRNA features (AUC of 0.9, and 95% CI: 0.75, 1.0). Models were validated with excellent performance in ML platforms and good performance in an independent dataset. Prediction models of EC recurrence containing lncRNA features have better performance than models with clinical data alone.

## 1. Introduction

Endometrial cancer (EC) is the most common gynecologic malignancy in developed countries. It is estimated that 65,950 new uterine cancer cases will be diagnosed in the United States in 2022, accounting for 12,550 deaths [1]. Unlike other cancer types, incidence and mortality of EC have been increasing for the last 2 decades [1]. This is mainly considered due to an aging population and increased rates of obesity and metabolic syndrome [2]. Obesity contributes to an endogenous unopposed estrogen environment and is the single most important risk factor for EC [2]. The increase in EC mortality has been projected to rise another 55% by 2030 due to the obesity epidemic [3].

In addition, over the last 2 decades the evidence from important clinical trials have changed standards of treatment for low-risk and low–intermediate-risk EC (PORTEC 1, and GOG 99) [4,5], high–intermediate-risk EC (PORTEC 2 and ASTEC) [6,7], and high-risk EC (PORTEC 3, GOG 249, and GOG 258) [8,9,10,11]. Despite those advances, treatment failure occurs in approximately 10–15% of patients with early stage EC. Although non-endometrioid variants, such as serous and clear cell carcinomas, comprise <10% of all diagnoses, they account for a disproportionately high number of EC recurrences and cancer-related deaths [12]. However, the majority of treatment failures and recurrences occur in endometrioid EC type (EEC) and prognosis remains poor for these women, with the exception of patients with isolated vaginal recurrence [12,13]. Thus, identifying patients who might benefit from additional surveillance and treatment to prevent recurrence and reduce mortality from this disease would be of great value.

The Cancer Genome Atlas (TCGA) identified molecular features that were found to categorize EC tumors into different levels of risk [14,15]. Later, the Post-Operative Radiation Therapy in Endometrial Carcinoma (PORTEC) Study Group included some of these molecular features to design its latest trial, 4a (NCT03469674). In this trial, standard adjuvant treatment with vaginal brachytherapy for women with high–intermediate-risk EC, is compared with individualized adjuvant treatment based on a molecular-integrated risk profile [16]. However, with this molecular assessment, almost 60% of patients presented a ‘no specific molecular profile’ (NSMP) [17]. Prior studies have also used clinical and pathological characteristics to stratify risk for recurrence and to inform adjuvant treatment [18,19]. Despite these studies, to date, there is no standard, validated, and accurate model that assesses individual risk of recurrence for patients with EC. Previous attempts reported accuracies between 60 and 73%, or area under the curve (AUC) around 80% [18,20]. Better models are needed to identify 15% of those patients with early stage EC that are going to need adjuvant treatment. 

We hypothesize that integration of clinical and genomic data will improve prediction models of recurrence in EEC. The objective of this pilot study is to create models with clinical and genomic data that will discriminate patients with EEC at risk of recurrence from disease. We validated these models in independent datasets (TCGA) and machine-learning analytical platforms.

## 2. Results

The flow of included patients are depicted in Figure 1 and included patients characteristics in Table 1. 

### 2.1. Creation of Prediction Models of EEC Recurrence

After RNA extraction, sequencing and analysis, we determined a series of genomics features that were used for the prediction analysis: (a) from the extracted transcriptome: gene, long non-coding RNA (lncRNA) and single exon expression; (b) we determined genomic variation, including single nucleotide variation (SNV), copy number variation by gene (CNV), and copy number variation by chromosomal region. Additionally, we identified structural variation (SV), including fusion transcripts (FT), retained introns (RI), novel exon/junction (NEJ), and unknown or previously not reported SV (UNK). After the univariate analysis with cross-validation of all genomic features, we found those characteristics that were more informative of EC recurrence (Figure 2). These significant features were later introduced in prediction models of recurrence. 

Next, we built prediction models for recurrence. Initially we constructed them with only one feature. Then, we made models with two and three different sets of variables. Adding four or more variables did not improve prediction models and added complexity to the system (Figure 3). In total, we built over 170 models to predict EEC recurrence (Appendix A). In Figure 3, we represented the 30 models with one or two types of variables with the best performance measured by AUC (Figure 3A), and the 30 best performing models with three types of variables (Figure 3B). If we consider that clinical data is the best way to assess risk of recurrence up to now, potentially superior models are those with a performance, measured by AUC, over 0.75, which is the basic clinical model performance (Figure 3A in lighter blue).

Notably, all best models included lncRNA data. Moreover, the model with only lncRNA had an AUC of 0.9 (95% CI 0.75–1.0) and adding more clinical or genomic data to the model did not improve the performance (Figure 3). No matter how many types of variables were added to lncRNA data, the multivariate regression lasso model ended up with the same five lncRNAs: ENSG00000274840, ENSG00000240137, ENSG00000250137, ENSG00000253622, and ENSG00000258240. So, comparing all models, the simplest model with only lncRNA turned out to have one of the best performances, with an AUC of 0.9, and the same final five variables at the end of the lasso analysis than more complicated models with more types of data (Figure 4). This simplest, best-performing model would be the one with potential to improve the only clinical model. 

### 2.2. Validation of Prediction Models of EEC Recurrence

Validation of the best performing model was performed with different analytical platforms and with an independent dataset (TCGA).

#### 2.2.1. Validation of Models with Machine Learning (ML)

We validated the best model with two different ML analytical platforms. The first one used *TensorFlow*, and we tested the model with (Figure 5A) and without (Figure 5B) FIGO Stage. The performance of both were excellent, with AUC of 1.00, and accuracies over 85%. For the second ML platform we used the suite MATLAB and its ML App. The App has over 30 ML methods than can be used in parallel to assess the accuracy of a model. Again, both AUC and accuracies were excellent (100%) for the model with lncRNA data and FIGO Stage (Figure 5C) and for the one with only five lncRNAs (Figure 5D). 

In summary, in this validation analysis, the best predictive model for EEC recurrence seems to be robust enough throughout different analytical methods and platforms. Additionally, adding clinical data (FIGO stage) to lncRNA data did not improve the performance of the validation model.

#### 2.2.2. Validation of Models with TCGA Dataset

Finally, we validated our best model with an independent, publicly available dataset, TCGA, with 406 EEC patients, 346 non-recurrent, and 60 recurrent. Patients’ characteristics of this dataset were similar to our study population and can be reviewed in the Appendix A. After extracting lncRNA data from the original BAM files, we selected the five lncRNAs that were driving the prediction model and tested them in the *TensorFlow* and MATLAB ML platforms. It had an accuracy of 78% and 86%, respectively, with also good AUC of 0.68 and 0.78 (Appendix A). Therefore, our best model also performed well in an independent dataset (TCGA).

## 3. Discussion

Our pilot study found that all best prediction models of EEC recurrence contained lncRNA features, and specifically five lncRNAs. The simplest, best performing model contained only five lncRNAs and was as accurate as more complex models. Furthermore, the lncRNA model was superior to the model with only clinical data (AUC of 0.9 versus 0.75, respectively) and with a 95% CI that reached 1.0. If these results were to be validated in future studies this would result in an accurate and robust model that could discriminate which EEC patients would be at risk of initial treatment failure at the time of surgery. This would leave healthcare professional with plenty of time to design alternative adjuvant treatments to prevent these outcomes.

As we hypothesized previously, integration of clinical and genomic data improves prediction models of EEC recurrence, with AUC performances of 0.9, and CI reaching 1.00. Integration of complex data is a difficult task, but the results could be very valuable [22,23]. It requires a constant dialogue between basic scientists, clinicians, statisticians and bioinformaticians to build models that are clinically and scientifically meaningful [24]; models that predict clinically significant outcomes that can be actioned upon. Our goal was to identify EEC patients at risk of recurrence who might benefit from additional surveillance and treatment to prevent relapse and reduce mortality from this disease. Our prediction models achieved that goal. 

As noted initially, EC trials incorporated pathological prognosticators to determine postoperative radiation, as initial attempts to individualized therapy [4,5,6,7,8,9,10,11,25]. However, treatment failures still seemed to burden low-risk patients determined by pathology, though the percentage was superior in high-risk women [12,13]. Molecular profiling of EC promised improvement in risk assessment and treatment selection, especially after the TCGA initiative [14,15]. TCGA described four groups that had different molecular features: POLE ultra-mutated, microsatellite-instability-hypermutated, copy-number-low, and copy-number-high EC. Later, these groups were modified to make the molecular determination more feasible and affordable—the Proactive Molecular Risk Classifier or *ProMisE* [26]. The resulting groups seemed to correlate well with disease prognosis [17,26], and have been used to design new EC trials [16]. However, several questions remain to be addressed. The first one is the high number of unclassifiable EC, as much as 59% by one of these studies [17], and whether pathology prognostic factors should be applied to these cases. Additionally, when reviewing the prediction performance of these molecular features, prediction of recurrence for all models varied from 60–75% [27], which are not superior to clinical models [18,20]. Recent ESGO/ESTRO/ESP guidelines for EC management [28] stated that molecular classification could impact clinical management, especially in cases with high-grade/high-risk disease. However, they recognized that the molecular classifier is not perfect, and there is room for improvement for those patients with low-risk and/or unclassifiable molecular features. This is where our prediction model could help, in EEC patients with seemingly low-risk disease and unclassifiable molecular features, which are the majority.

LncRNAs have regulatory functions [29,30]. They participate in epigenetic regulation, maintain chromatin structure, and modulate transcription [30,31]. There is increasing reporting of the function that lncRNA have in the development and progression of EC [32]. The development of EC is a complicated biological process and lncRNAs may act as oncogenes or tumor suppressors. Their expression may contribute to EC transformation and the subsequent progression. Gene expression experiments have previously demonstrated that a large number of lncRNA expression is altered in EC [33]. Therefore, it is not surprising that some lncRNAs were selected in the univariate analysis because of their difference in expression between recurrent and non-recurrent samples. Furthermore, some of the lncRNAs present in our best model have been described previously in several cancers, ENSG00000274840, ENSG00000240137, and ENSG00000253622 [34,35,36], and ENSG00000250137 have been associated with increased BMI [37]. Our best model may be reflecting the underlying biological characteristics of the recurrent EEC phenotype. 

The strengths of our study include our use of a comprehensive genomic characterization of EEC samples, including genomic variation and structural variation to determine the best prediction model of EEC recurrence. Additionally, we used proven statistical methods that employed internal validation with cross-validation for feature selection to avoid model over-fitting. Finally, we performed external validation with diverse analytical platforms, including the use of ML, and different and independent datasets of EEC (TCGA) analyzed with identical methods and software, also described previously [38,39]. It should be mentioned that genomic variation, specifically SNV and CNV, are better determined with DNA sequencing. Extracting genomic variation from RNAseq is an estimation of the real somatic variation (~75% of the variants) [40,41], but it served the purpose of this study and prevented cost scalation. In the end, no SNV or CNV were in the best model of EEC recurrence prediction. 

One of the potential limitations of the study is the relativity low number of recurrent EEC that have complete clinical and genomic data in our dataset. Recurrence of EEC range between 10–15% [12], and even larger genomic databases, such as TCGA, have a low and unbalanced number of recurrent cases that may affect any prediction model. To adjust for this low and unbalanced number of recurrent cases, we validated all models with ML analytics that specifically account for this issue by resampling data during model training, validation, and testing [42]. Our best model is the result of a well-studied EEC population [43,44,45,46] that has been well annotated and followed over the years. We are a state-sponsored University, which receives and serves the vast majority of patients with gynecological cancer in the State of Iowa. However, as the population of Iowa is predominantly white, 92% of patients included in the study were white. The lack of diversity of our selected subjects is a potential limitation of our analysis that may influence the generalizability of the study to other states where there is more diversity. In a previous study comparing our population with TCGA population, we identified differences in the admixture of both cohorts [46]. These differences may have an effect on the performance and validation of the model outside Iowa. Other limitation of the study may arise from inherited biases of retrospective studies, mainly recall biases. Due to losses in follow-up, some of the recurrences could be under-reported. However, we are more concerned with the surveillance of TCGA EEC patients that may influence the performance of validation analysis. Finally, we have to be aware of potential overfitting of our recurrence prediction model, either to our own population and/or to our own data. To avoid this issue, the performance of the model has to be confirmed in a new prospective set of diverse EEC patients where the phenotype is known confidently. Until then, the model should not be used clinically.

## 4. Materials and Methods

### 4.1. Study Design

We performed a retrospective, single institution, case–control study in which we included 62 patients with EEC from 1991 to 2010 available in our biobank with pre-operative and intra-operative clinical data. RNA was extracted from tumor specimens and processed as detailed below to obtain the necessary genomic data. Clinical and genomic data were then combined to create predictive models using statistical learning to identify criteria which accurately predicted recurrence for EC patients.

### 4.2. Ethics and Tissue Procurement

Tissue samples and clinical outcome data were obtained from the Department of Obstetrics and Gynecology Biobank (IRB, ID#200209010), which is part of the Women’s Health Tissue Repository (WHTR, IRB, ID#201804817). All tissues archived in the Gynecologic Oncology Biobank (herein termed Biobank) were originally obtained from adult patients under informed consent in accordance with University of Iowa IRB guidelines. Tumor samples were collected, reviewed by a board-certified pathologist, flash-frozen, and then the diagnosis was confirmed in paraffin. All experimental protocols were approved by the University of Iowa Biomedical IRB-01. 

### 4.3. Clinical Data Procurement

Clinical data was extracted from the electronic medical record. Table 1 summarizes the baseline clinical and pathologic characteristics. Only data that were available by the end of initial treatment were used in the development of predictive models. Pre-operative characteristics included age at diagnosis, body mass index (BMI), pre-operative hemoglobin, serum creatinine, albumin, comorbidities (coronary artery disease, diabetes mellitus, congestive heart disease, history of cardiovascular accident, tobacco use), and Charlson morbidity index. Intraoperative characteristics included type of surgery (laparoscopic, robotic, laparotomy, vaginal), operative time, and estimated blood loss. Post-operative characteristics extracted included final pathology diagnosis, disease stage, estrogen and progesterone receptor status, surgical complications, adjuvant therapy (including types of radiation therapy), and recurrence of disease. 

For the purposes of this study, we defined disease recurrence as EEC diagnosed in any location after completing treatment and a subsequent period with no evidence of disease. Of the 62 EEC patients with clinical and genomic information included in the study, 7 had a recurrence of disease. Two patients had an advanced stage with persistence of disease after initial treatment and, by the study definition, they were considered as non-recurrent. All patients recurred within 5 years; 86% of them experienced a recurrence within 2 years of initial treatment. Differences between clinical variables among both study groups were assessed by logistic regression (significance at *p*-value < 0.05).

### 4.4. Genomic Analysis

#### 4.4.1. Included Subjects

A cohort of 127 patients diagnosed with EEC at UI was assembled under approval by the Institutional Review Board (IRB# 201607815). Only patients with a confirmed EEC diagnosis, with clinical follow-up and biological specimens with quality RNA for sequencing, were included in the study. The flow diagram in Figure 1 summarizes patients included in this study. 

#### 4.4.2. RNA Isolation and Sequencing

RNA was then isolated from these tumor specimens. RNA extraction, processing and sequencing have been described previously [39,43]. In brief, total cellular RNA was extracted from primary tumor tissue using the mirVana (Thermo Fisher, Waltham, MA, USA) RNA purification kit. The RNA yield and quality were assessed with Trinean Dropsense 16 spectrophotometer and Agilent Model 2100 bioanalyzer. RNA quality was determined to be adequate if the sample had an RNA integrity number (RIN) of 7.0 or greater. Samples that were of adequate quality were then sequenced. 500ng of RNA was quantified by Qubit measurement (Thermo Fisher). RNA was then converted to cDNA and ligated to sequencing adaptors with Illumina TriSeq stranded total RNA library preparation (Illumina, San Diego, CA, USA). cDNA samples were then sequenced with the Illumina HiSeq 4000 genome sequencing platform using 150 bp paired-end SBS chemistry. All sequencing was performed at the Genome Facility at the University of Iowa Institute of Human Genetics (IIHG). 

#### 4.4.3. Data Preprocessing

*STAR* was used to align the RNAseq reads to the human reference genome (version hg38) [47]. We then created BAM files after alignment. *FeatureCount* was used to measure gene expression [48]. The *DESeq2* package was used to import, normalize, and prepare the gene counts for analysis [49]. *ENSEMBL* was used to annotate single exons within the gene expression alignment analysis. Exon expression was then evaluated using the *DEXSeq* package [50]. BAM files for each sample were used to estimate SNV discovery and base-calling against the human genome reference utilizing *SAMtools* and *BCFtools* for sorting and indexing. Results were filtered for duplicates, known non-synonymous single-nucleotide variants, and synonymous variants and then annotated with *ANNOVAR*. Gene CNV were estimated using *SAMtools* and *superFreq* [40]. BAM files were then used to identify lncRNA, as described previously [51,52]. Lastly, fusion transcripts were determined using the *STAR-Fusion* suite from fastq files [53]. Appendix A depicts each program used for RNA processing and the identification of various genomic components. 

### 4.5. Statistical Analysis

Most genomic data were used as continuous variables, except SNV and structural variation features, which were used as dichotomous variables. To select those variables most informative for prediction of EEC recurrence, we used univariate analysis with ANOVA (*p* < 0.05) and cross-validation with 10 replicates for each fold, as implemented by the *caret* R package and detailed previously [39]. Significant predictive variables were then used in a multivariate *lasso* regression prediction model (statistical learning). Thus, poorly annotated variables were removed from model construction. 

#### 4.5.1. Creation of Prediction Models of EEC Recurrence with Statistical Learning

Significant variables from the univariate analysis were then incorporated into multivariate *lasso* (least absolute shrinkage and selection operator) regression prediction models of recurrence. Initial models included only significant variables from one category of clinical or genomic data (i.e., lncRNA expression, gene expression, CNV, etc.). Variables were then progressively combined to create more complex prediction models. Multivariate prediction models were fit with *lasso* as implemented in the *glmnet* R package [54], and detailed previously [38,39]. Performances of prediction models were measured with area under the receiver operating characteristics curve (AUC) and their respective 95% confidence intervals (CI) and estimated with 1000 replicates of ten-fold cross-validation to avoid over-fitting. Bias-corrected and accelerated bootstrap CIs were computed for each model. AUC of 0.5 indicates no predictive ability of a model and 1.0 represents perfect predictive performance.

#### 4.5.2. Validation of Predictive Models with Machine-Learning Methods

For validation of the best prediction models of recurrence in a machine-learning platform we used *TensorFlow* [55] in a *Jupyter* notebook with a *Keras* application programming interface (API) [42]. *TensorFlow* code was modified from a tutorial (found here: https://www.tensorflow.org/tutorials (accessed on 25 October 2022)). Training, validating, and testing were performed to account for weights of the outcomes as well as for unbalanced data (mainly for complete vs. optimal patients). Additionally, we validated the best prediction models in *MATLAB* machine learning app, where there are over 20 classifier methods. Model from UI were validated in this new analytical platform and later was validated in EEC TCGA dataset. 

#### 4.5.3. Validation of Predictive Models with Independent Data, TCGA

Data from TCGA dataset for endometrial EC were downloaded from the National Cancer Institute (NCI) database in accordance with TCGA Human Subject Protection and Data Access Policies, adopted by the NCI and the National Human Genome Research Institute (NHGRI). Data were downloaded with the NCI database of genotypes and phenotypes approval (dbGaP#16003) as previously described [43]. Patients with non-endometrioid histology were excluded. Clinical and molecular data were obtained from 406 patients diagnosed with EEC, of which 60 experienced recurrence of disease as defined above (Appendix A. Original downloaded BAM files were then used to identify lncRNAs, as described previously [51,52].The best-performing parameters were used to fit a final score of that model to the entire TCGA cohort [39]. Performances measured by AUC between 0.8–0.9 were considered ‘very good’; performances between 0.9–1 were considered ‘excellent’.

## 5. Conclusions

Prediction models containing lncRNA features have better performance, measured by AUC, than models with clinical data alone. These models must be validated in prospective manner and different populations before their use in clinical settings.

## Figures and Tables

**Figure 1 ijms-23-16014-f001:**
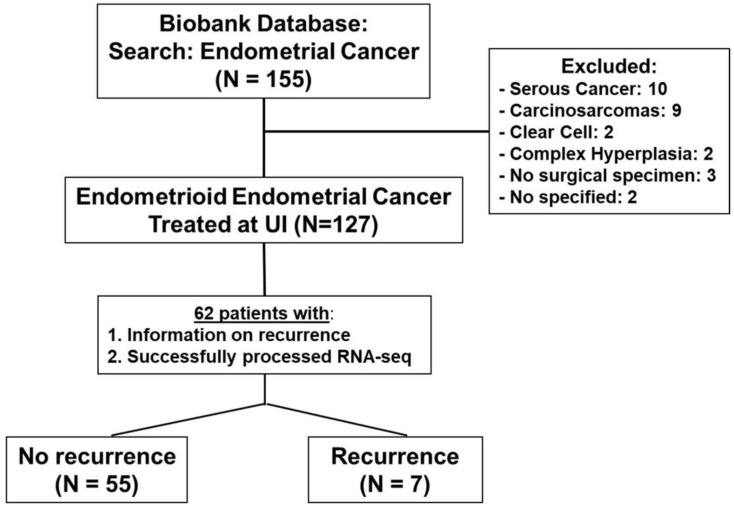
Flow of included patients in analysis. Of the initial 155 endometrial cancers available in the UI Biobank, 127 were confirmed to be of endometrioid histology. The rest were excluded from the study. A total of 62 patients had annotated follow-up with detailed recurrence information and with quality RNA for RNA sequencing (RNA-seq).

**Figure 2 ijms-23-16014-f002:**
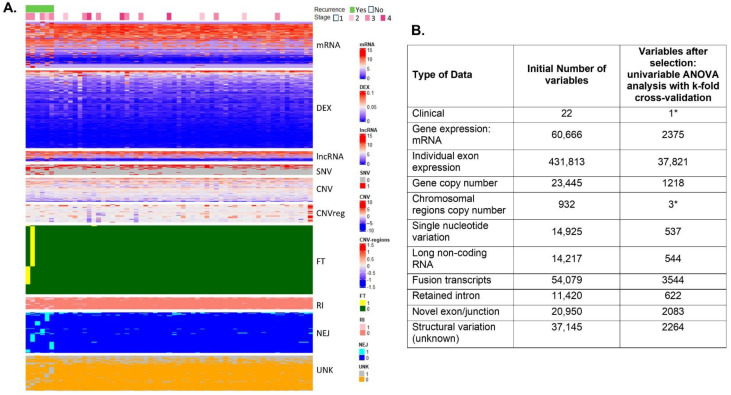
Included patients. (**A**) Heatmap of selected variables after univariate ANOVA analysis. Representation of the significant variables after univariate analysis (*p* < 0.05) for different types of genomic data. Recurrent cases are at the left side of the panel (under the light green bar); non-recurrent cases are at the right (white bar). Of the 22 clinical features introduced in the lasso analysis, only stage was informative for recurrence (upper part of the panel, color coded from 1 to 4). Transcriptome: DEX: exon expression; lncRNA: long non-coding RNA; Exp: gene expression. Genomic variation: SNV: single nucleotide variation; CNV: gene copy number by gene; CNVreg: copy number by chromosomal region; Structural variation: FT: Fusion transcripts; RI: Retained intron; NEJ: Novel exon/junction; U-SV: Unknown SV. At the right side of the panel are the labels and the color-coded range of values for all genomic variables. (**B**) Variable selection and variables after univariate analysis. To reduce the number of variables, we used univariate analysis of all data with ANOVA to select the variables that were more informative for prediction of response, with a *p*-value < 0.05 (3rd column). * Lasso regression was performed directly with no pre-reduction with ANOVA because the smaller number of variables in two types of data: clinical data and copy number by chromosomal region. Graphics were generated with R package *ComplexHeatmap* [21].

**Figure 3 ijms-23-16014-f003:**
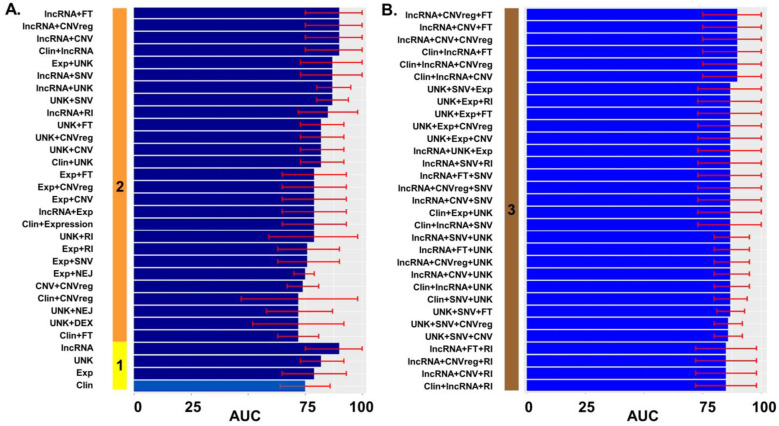
Performance of prediction models of EEC recurrence. (**A**) The solid vertical bar represents the number of types of data: 1 (yellow): only one variable was included in the model; 2 (orange): combination of two types of variables; (**B**) The solid vertical maroon bar represents the combination of three types of variables. Different performances on both panels are displayed in ascending order. The *x* axis is AUC as a percentage (0–100%). The red error mark displays the 95% confidence interval (CI). Overall, 72 models with different combinations of data were tested. We only displayed the best (**A**) 30 models for combinations of one and two variables and (**B**) 30 best models for combinations of three types of variables. Exp: gene expression; DEX: exon expression; lncRNA: long non-coding RNA; SNV: single nucleotide variation; CNV: gene copy number by gene; CNVreg: copy number by chromosomal region; FT: Fusion transcripts; RI: Retained intron; NEJ: Novel exon/junction; UNK: Unknown SV. Graphics were generated with R package *ggplot*.

**Figure 4 ijms-23-16014-f004:**
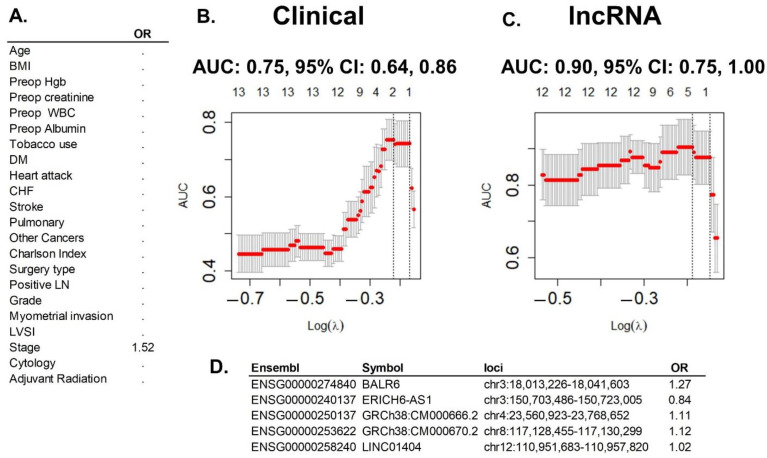
Best prediction model and comparison with clinical model. (**A**) The lasso multivariate regression model with clinical variables: out of the two clinical variables included in the model, only Stage remained as informative for prediction of EEC recurrence after the analysis, with an increased risk for recurrence as stage increases. (**B**) Graphic representation of the clinical lasso analysis: superior margin reflects number of variables; left margin reflects performance of the model measured in AUC (area under the curve); lower margin reflects lambda tunning parameter chose by cross-validation to optimize the model. The optimized AUC was 0.75 (95% CI: 0.68, 0.86), between the dotted lines. (**C**) Graphic representation of the lncRNA data lasso analysis (same margins and design as before): optimized AUC of 0.9 (95% CI: 0.75, 1.00). (**D**) The lasso multivariate regression model with lncRNA data: out of the 544 clinical variables included in the model, five single lncRNAs remained as informative for prediction of EEC recurrence. Four of them increased risk (OR > 1) and one protected from recurrence (OR < 1). Graphics were generated with R package glmnet.

**Figure 5 ijms-23-16014-f005:**
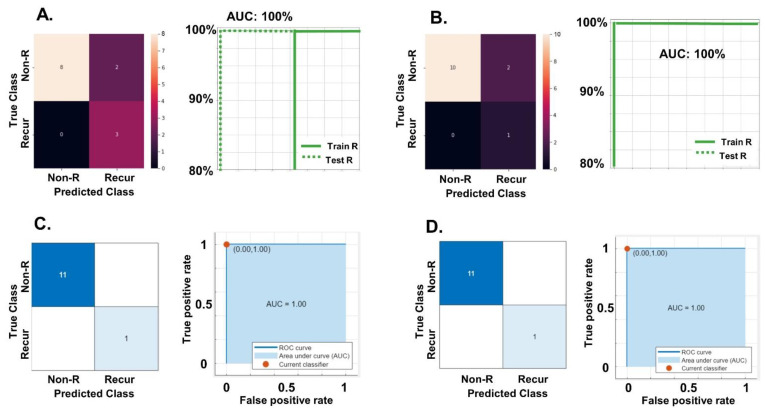
(**A**) Validation of the best model with lncRNA data and FIGO Stage clinical variable (the most informative in the clinical model). These are the results after training and validating the model with TensorFlow in 75% of the samples, and then performing testing on the remaining 25% of the data. We are showing the testing results. The left panel shows the confusion matrix representing the true (True Class) versus the predicted values (Predicted Class). The right panel is an ROC graphic: true positives in the *x* axis, false positives in the *y* axis, and AUC results. Train R: results of unbalanced (or re-sampling) model training; Test R: results of re-sampling model testing. AUC of 1.00 and accuracy of 0.92. Recur: recurrent; Non-R: non recurrent. (**B**) Validation of the best model with only lncRNA data and no clinical variables. As before, we are showing testing results in 25% of the data, after training and validating have been performed with TensorFlow. Left and right panels are as before. AUC of 1.00 and accuracy of 0.85. (**C**) Validation of the best model with lncRNA data and FIGO Stage clinical variable performed with MATLAB platform. We are showing testing results in 20% of the data, after training and validating have been performed. MATLAB offers over 30 methods for its machine learning (ML) App. In four of them the accuracy of testing was 100%, as shown in the graphic. Specifically, this is the coarse Gaussian SVM (support vector machines) method. Left and right panels are as before. AUC of 1.00 and accuracy of also of 1. (**D**) Validation of the best model with only lncRNA data and no clinical variables with MATLAB platform. Parameters are as before. This time, most methods had a testing accuracy of 100%. Showing the linear SVM method. Left and right panels are as before. AUC of 1.00 and accuracy of also of 1.

**Table 1 ijms-23-16014-t001:** Clinical patient characteristics. These are the baseline variables determined at treatment completion and included in the analysis.

		Recurrent (N = 7)	Non-Recurrent (N = 55)	*p*-Value
**Age**	(average)	61	61	0.983
**BMI**	(average)	35.2	37.1	0.627
**Charlson Index ****				0.720
	Low (1–3)	0	9	
	Medium (4–6)	7	37	
	High (>6)	0	5	
**Personal History**	DM	1	13	0.582
	Heart atack	0	1	0.995
	CHF	0	1	0.995
	Stroke	0	2	0.996
	Pulmonary disease	0	8	0.994
	Other cancers	0	8	0.994
**Grade**				0.731
	1	2	23	
	2	4	20	
	3	1	10	
**Lymphovascular involvement**				0.208
	No	4	42	
	Yes	3	11	
**MI**	(average)	69.3	36.6	0.029 *
**Cytology**				0.450
	No	5	50	
	Yes	1	4	
**Stage**				0.009 *
	I	1	43	
	II	1	3	
	III	5	6	
	IV	0	3	
**Adjuvant Radiation (any type)**				0.150
	No	3	39	
	Yes	4	16	

Tobacco use, pre-operative Hgb, creatinine, WBC, albumin, type of surgery -open or minimally invasive-, surgical complications (including blood loss), and length of stay were not significantly different. BMI: body mass index; MI: myometrial invasion. * Statistically significant with *p*-value < 0.05. ** Charlson Comorbidity Index is a measure of the prognostic burden of all associated morbidities to predict mortality and is the most validated measure of the prognostic impact of multiple chronic illnesses (www.charlsoncomorbidity.com (accessed on 30 July 2022)).

## Data Availability

Clinical data are not publicly available due to patient privacy. Datasets can be browsed by their accession number: GSEXXXXX (pending). The validation part of this study was performed in silico, with de-identified publicly available data. All data from TCGA is available at their website: https://portal.gdc.cancer.gov/, accessed 5 December 2022. Software utilized by this study is also publicly available at Bioconductor website: http://bioconductor.org/, accessed 3 October 2022.

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
