# Peer review of "Integration of Genomic and Clinical Retrospective Data to Predict Endometrioid Endometrial Cancer Recurrence"

_ijms, 2022, doi:10.3390/ijms232416014_

Round 1

Reviewer 1 Report

Gonzalez-Bosquet and colleagues presented a research article aimed at identifying novel prognostic indicators to predict the risk of recurrence in endometrial cancer. For this purpose, the authors performed genomic and clinical investigations on a pilot cohort of EC patients with or without recurrence. Although the research idea is interesting and the methodologies used are quite satisfactory, there are some issues that significantly limit the impact of the study. Please address the minor/major revisions listed below:

1) I have some doubts related to the low number of patients with tumor recurrence. You included only 7 patients with recurrence, of which 5 were stage III EC. The data generated by RNAseq in a small cohort of EC patients may generate biased and confounding data. Please argue this issue;

2) In chapter 2.2.2. please indicate, at least, the number of recurrent and non-recurrent EC analyzed;

3) Please indicate in the main manuscript the 5 lncRNAs identified (maybe in chapter 2.2.2.);

4) In the Introduction or Discussion section, the authors have to briefly mention how a multidisciplinary approach could be helpful for the management of gynecological malignancies. In particular, it was well-established as a multi-omics approach to gynecological cancer may be useful for the early diagnosis and the prediction of prognosis of both endometrial and ovarian cancer. In your study, you concomitantly analyzed clinical and molecular data to obtain more reliable prognostic indications, thus highlighting how a multidisciplinary approach may improve the clinical management of gynecological patients. Please add a brief description of this topic. For this purpose, please see:

- PMID: 34132354

- PMID: 35163161

- PMID: 30364388

5) In the following sentence please confirm that “Gynecology Gynecologic” is correct: “Tissue samples and clinical outcome data were obtained from the Department of Obstetrics and Gynecology Gynecologic Oncology Biobank (IRB, ID#200209010), which is part of the Women’s Health Tissue Repository (WHTR, IRB, ID#201804817).”; 

6) Please remove Panel B from Figure 1 and provide the Clinical data of patients as a new Table.

Author Response

Comments and Suggestions for Authors

Gonzalez-Bosquet and colleagues presented a research article aimed at identifying novel prognostic indicators to predict the risk of recurrence in endometrial cancer. For this purpose, the authors performed genomic and clinical investigations on a pilot cohort of EC patients with or without recurrence. Although the research idea is interesting and the methodologies used are quite satisfactory, there are some issues that significantly limit the impact of the study. Please address the minor/major revisions listed below:

1) I have some doubts related to the low number of patients with tumor recurrence. You included only 7 patients with recurrence, of which 5 were stage III EC. The data generated by RNAseq in a small cohort of EC patients may generate biased and confounding data. Please argue this issue;

We have added an argument about this point to the last paragraph of the Discussion section that comments on limitations of our study.

2) In chapter 2.2.2. please indicate, at least, the number of recurrent and non-recurrent EC analyzed;

Added in chapter 2.2.2

3) Please indicate in the main manuscript the 5 lncRNAs identified (maybe in chapter 2.2.2.);

They are included in the chapter ‘2.1. Creation of prediction models of EEC recurrence’, Page 5, before Figure 4 where the model is represented.

4) In the Introduction or Discussion section, the authors have to briefly mention how a multidisciplinary approach could be helpful for the management of gynecological malignancies. In particular, it was well-established as a multi-omics approach to gynecological cancer may be useful for the early diagnosis and the prediction of prognosis of both endometrial and ovarian cancer. In your study, you concomitantly analyzed clinical and molecular data to obtain more reliable prognostic indications, thus highlighting how a multidisciplinary approach may improve the clinical management of gynecological patients. Please add a brief description of this topic. For this purpose, please see:

- PMID: 34132354

- PMID: 35163161

- PMID: 30364388

We added a second paragraph in the Discussion section about interdisciplinary collaboration in the multi-omics approach for outcome prediction, including the references suggested by the reviewer.

5) In the following sentence please confirm that “Gynecology Gynecologic” is correct: “Tissue samples and clinical outcome data were obtained from the Department of Obstetrics and Gynecology Gynecologic Oncology Biobank (IRB, ID#200209010), which is part of the Women’s Health Tissue Repository (WHTR, IRB, ID#201804817).”; 

Gynecologic Oncology was deleted from the paragraph.

6) Please remove Panel B from Figure 1 and provide the Clinical data of patients as a new Table.

The table is now presented independently (Table 1)

Reviewer 2 Report

The presented paper is interesting for readers and have a scientific potential. Although I have some critical comments to the authors which help to improve their paper.

1. text in the figure 1B should be present as a table.

2. please check once again if all key words come from Mesh database.

3. the title should be more informative and includes information about the nature of this study - retrospective.

4. methods section should be split to smaller one - ethics, study design, subjects, RNA isolation, sequencing, miRNA analysis etc.

5. please add limitations and strengths of this study.

6. all abbreviations have to be defined at the first time.

7. lines 223-226 should not be used in the academic English. Please revise whole paper and eliminate errors.

Author Response

Comments and Suggestions for Authors

The presented paper is interesting for readers and have a scientific potential. Although I have some critical comments to the authors which help to improve their paper.

  1. Text in the figure 1B should be present as a table.

The table is presented now independently as Table 1.

  1. Please check once again if all key words come from Mesh database.

Key words were changed based on Mesh database.

  1. The title should be more informative and includes information about the nature of this study - retrospective.

As suggested, we added ‘retrospective’ into the Title describing the nature of clinical and genomic data.

  1. Methods section should be split to smaller one - ethics, study design, subjects, RNA isolation, sequencing, miRNA analysis etc.

Done

  1. Please add limitations and strengths of this study.

The last 2 paragraphs of the Discussion section have been dedicated to strengths and limitations of the study now.

  1. All abbreviations have to be defined at the first time.

We apologize for the confusion. One of the Figure legends (Figure 3) leaked into the text and created confusion. It has been corrected. We now have reviewed the manuscript to ensure the abbreviations are defined at the first time.

  1. lines 223-226 should not be used in the academic English. Please revise whole paper and eliminate errors.

Apologize for the colloquialism. It has been fixed. Also we have reviewed the rest of the text.

Round 2

Reviewer 2 Report

The paper has been improved according to the Reviewers' suggestions.